# Designing Cropping Systems to Improve the Management of the Invasive Weed *Phalaris minor* Retz.

**Gaofeng Xu** [1,2], **Shicai Shen** [2], **Yun Zhang** [3], **David Roy Clements** [4], **Shaosong Yang** [2], **Jun Li** [1], **Liyao Dong** [1,*], **Fudou Zhang** [2,*], **Guimei Jin** [2] and **Yuan Gao** [1]

[1] College of Plant Protection, Nanjing Agricultural University, Nanjing 210095, China; xugaofeng1059@163.com (G.X.); li_jun@njau.edu.cn (J.L.); yuangao@njau.edu.cn (Y.G.)
[2] Institute of Agricultural Environment and Resources Research, Yunnan Academy of Agricultural Sciences, Kunming 650205, China; sshicai@csu.edu.au (S.S.); yshaos@163.com (S.Y.); yfjgm2019@163.com (G.J.)
[3] Biotechnology and Germplasm Resources Institute, Yunnan Academy of Agricultural Sciences, Kunming 650205, China; zhangyun507@163.com
[4] Biology Department, Trinity Western University, 7600 Glover Road, Langley, BC V2Y 1Y1, Canada; clements@twu.ca
[*] Correspondence: dly@njau.edu.cn (L.D.); fdzh@vip.sina.com (F.Z.); Tel.: +86-256-692-8708 (L.D.); +86-871-6589-4429 (F.Z.)

**Abstract:** Because cropping systems can greatly affect the establishment and spread of alien species populations, the design of cropping systems to control invasive weeds is an important approach for invasive species management in agro-ecosystems to avoid excessive increases in other control measures such as herbicides. The annual weed *Phalaris minor* Retz. (*P. minor*) is one of the most troublesome invasive weed species of winter crops in Yunnan Province, China, but the development of cropping systems for ecological control of this weed have received limited research attention. Here, we studied seed dormancy, germination characteristics and reproductive responses of *P. minor* to various cropping systems to show how cropping systems could be better designed to control *P. minor* in China. Our research showed that cropping systems significantly affected seed dormancy in submerged paddy fields. *Phalaris minor* seed remained dormant and the germination rates (less than 10%) were significantly lower ($p < 0.05$) than in maize fields and dry, bare soil conditions. Wheat, faba bean and rapeseed crops had no significant influence ($p < 0.05$) on the seed germination rate of *P. minor*, but increasing soil depth significantly decreased ($p < 0.05$) the germination rate and germination index of this weed. Total biomass, spike biomass, spike number and seed number of *P. minor* were significantly reduced ($p < 0.05$) with increasing proportions of the three crops (wheat, faba bean and rapeseed), with rapeseed having the strongest inhibition effects among the three crops. The reproductive allocation and reproductive investment of *P. minor* were also significantly reduced ($p < 0.05$) in mixed culture with wheat and rapeseed. With increasing proportions of wheat or rapeseed, the specific leaf area of *P. minor* significantly increased ($p < 0.05$), but the reverse was true for leaf area and specific leaf weight. Moreover, the net photosynthetic rate, stomatal conductance and transpiration rate for *P. minor* also decreased significantly ($p < 0.05$) when grown with wheat or rapeseed. These results suggest that optimal cropping systems design could involve planting rapeseed in conjunction with deep plowing and planting rice (continuous submergence underwater) in summer. Such a system could reduce the field populations and seed bank of *P. minor*, thus providing a sustainable and environmentally friendly means of suppressing *P. minor*.

**Keywords:** *Phalaris minor*; cropping systems; ecological control; seed dormancy; seed germination; reproductive characteristics

## 1. Introduction

*Phalaris minor* Retz. (*P. minor*), littleseed canarygrass (Poaceae), is an annual weed native to the Mediterranean region [1,2]. Currently, it has been reported in more than 60 countries and is widely distributed in every continent except for the polar regions [2,3]. The invasive weed, *P. minor*, was considered to be the predominant and most troublesome annual grass weeds in agro-ecosystems in many regions, and has resulted in serious yield reductions of most agronomic crops, particularly wheat [4,5]. At present, use of chemical herbicides (e.g., isoproturon, fenoxaprop-P-ethyl and pinoxaden) is the main method used for control of this weed [2,6,7]. However, long-term usage of chemical herbicides has created environmental pollution, herbicide resistance and human health problems [8]. Moreover, use of chemical herbicides and/or manual removal are difficult and have low efficiency for controlling *P. minor* at the wheat seedling stage because the two species have similar morphology, physiology and phenophases [2,9,10]. Utilizing crop allelopathy to inhibit this weed is one of the more environmentally friendly approaches tested, but few have succeeded [11,12]. Controlling this weed using ecologically sustainable approaches is still a worldwide challenge.

The successful establishment of non-native species is closely related to the invaded habitats [13,14]. For agro-ecosystems, modifying cropping systems could affect the occurrence and damage caused by invasive species, potentially affecting their control strategies and control efficiency [15,16]. Therefore, it is essential to determine the spatial-temporal population dynamics of *P. minor* in different cropping systems, which is also an important prerequisite for controlling this weed. As an annual herbaceous invader, the seed and seedling characteristics of *P. minor* play an important role in successful establishment, and therefore modification of cropping systems could affect seed dormancy, germination and reproduction. Thus, studies of the bio-ecological characteristics of seed dormancy, germination characteristics and reproductive responses to various cropping systems are critical for future ecological control of *P. minor*.

Previously, information on seed dormancy, germination and reproductive characteristics of *P. minor* showed that after dispersal *P. minor* seeds underwent true dormancy for 3–6 months and seed dormancy could be regulated by environmental factors [17,18]. For example, the germination of *P. minor* seed was higher when the seed was exposed to white light after dispersal compared to germination in the dark (wrapped in aluminum foil) [18]. Moreover, studies demonstrated that this weed generally produced 10–50 spikes per plant and generated a large number of seeds in wheat fields. Some studies have also shown that wheat varieties and planting density affected *P. minor* seed production [19,20].

*Phalaris minor* became established in China in the 20th century and is presently among the most destructive invasive weeds in temperate and subtropical regions of Yunnan Province, China [21,22]. In these regions, wheat, faba bean and rapeseed are the main winter crops, and rice and maize are the main summer crops. In a previous study, we investigated the population characteristics of *P. minor* in different crop types and found the population density of *P. minor* in the field with long-term planting of rapeseed were significantly lower than when planting wheat or faba bean [22]. In subsequent studies, the measurement of the total biomass and tiller number of *P. minor* indicated that rapeseed was more competitive than wheat and faba bean in suppressing *P. minor* [23]. However, there was still limited information available on how particular crops and cropping systems might influence the occurrence and damage of *P. minor*.

In our previous studies, the population characteristics of this weed were reported for different crop types. However, the biological and ecological characteristics of various cropping systems are not well characterized. For the present study, we set up greenhouse and field experiments to investigate seed dormancy, germination characteristics and reproductive response of *P. minor* to various crop types and tillage methods. Specifically, our objective was to test the hypothesis that cropping systems could be designed to improve the control of the invasive weed *P. minor* in agro-ecosystems.

## 2. Materials and Methods

### 2.1. Study Site

Field experiments were conducted at Songming Experimental Base of Yunnan Academy of Agricultural Sciences, Kunming city (25°12′–25°39′ N, 102°76′–102°89′ E). This area is characterized by a typical north-temperate climate, rainfall averages 1035 mm per year and the annual mean temperature is 15 °C [24]. The experiment site is in a rice-wheat rotation and no herbicide was used in the previous crop. The soil properties were: pH 6.2, organic content 18.14 ± 0.17 g/kg, total N 1.63 ± 0.12 g/kg, total P 0.37 ± 0.08 g/kg, and total K 8.41 ± 0.17 g/kg. Although *P. minor* is now widely distributed in farmland of Kunming City, there was no *P. minor* observed at the experimental site over the past 2 years.

### 2.2. Study Species

*Phalaris minor* is one of the most serious invasive species in winter crops, causing yield losses in wheat and other winter crops, as seen in the Kunming area, where this study took place [5]. Seeds of *P. minor* were collected from wheat fields near the field test sites on 15 April 2017. The average weight of 1000 seeds was 1.49 ± 0.05 g and the germination rate was 91.8%, as tested prior to experimentation.

Rice, maize, wheat, faba bean and rapeseed are the main summer and winter crops in Yunnan Province. Rice variety (cv. Yunjing No. 36), maize variety (cv. Yunrui No. 88), wheat variety (cv. Yunxuan No. 2), broad bean variety (cv. Fengdou No. 6) and rapeseed (cv. Yunyou No. 2) were obtained from the Rice Research Institute, Yunnan Academy of Agricultural Sciences (YAAS) for use in the experiments.

### 2.3. Influence of Cropping Systems on Seed Dormancy of P. minor

To understand the effect of cropping systems on seed dormancy of *P. minor*, a simulation test in the greenhouse was performed in 2017 from April to August, where average temperatures were between 20 °C and 35 °C. The experiment consisted of three treatments: (A) Seeds buried in dry soil (never watered and no plants present); (B) seeds buried in a maize field (watered once every 5 days); (C) seeds buried in a rice field (kept continuously submerged in water). Each treatment was replicated four times in a randomized block design. All of the plots were placed in separate cement pools (1.5 × 1.5 m), for a total of 12 cement pools. We selected 200 seeds with uniform size and color, put them into a nylon fabric attached to a long rope. We buried 4 nylon fabric bags randomly within of each plot at 12 cm depth, and kept the long rope on the surface. Then, we planted rice and maize in the corresponding plots at a constant planting density of 36 plants m$^{-2}$ (0.20 × 0.20 m spacing) on 10 April. During the experiment, no fertilizers or pesticides were used and the weeds within the rice and maize plots were removed manually except for *P. minor*. Every month (up to 5 months), we randomly took out a seed bag from each plot, collected 100 seeds and cleaned. Seeds were separated from the medium and sown in Petri dishes (four replicates) on different dates and kept at 20 °C in an incubator. During the experiment, seedling emergence was recorded every day. A seed was coded as "germinated" if the new shoots exceeded 1 cm in length.

### 2.4. Influence of Cropping Systems on Seed Germination of P. minor

To understand the effect of the cropping system on seed germination of this weed, we tested the seed germination rate of rhizosphere soil for different crops at different depths on 25 September 2017. We planted wheat, faba bean and rapeseed in plastic nursery seedling trays (54 × 28 cm), which had 50 holes, in the greenhouse on 8 October 2017. After seeding, we moved the plastic nursery seedling trays to cement pools with a water depth of 1–2 mm and maintained one plant in each hole by thinning. Rhizosphere soil samples for wheat, faba bean and rapeseed were collected 30 days after sowing. Soil without planted crops was treated as the control.

A potted plant experiment was carried out to investigate the effects of different crops and soil depths on seed germination of *P. minor*. The experiment consisted of four different soils, including wheat rhizosphere soil, faba bean rhizosphere soil, rapeseed rhizosphere soil and control soil, combined with

five treatments of burial depth, including 0, 3, 6, 9 and 12 cm. Plastic containers ($50 \times 20 \times 16$ cm deep) were filled with 14 cm deep rhizosphere soil of wheat, faba bean, rapeseed and control soil, respectively. Each container was further divided into five equal boxes ($10 \times 20 \times 16$ cm deep) with each randomly assigned to 1 of 5 burial depth treatments, and in each box, 5 gaps with the same depth were extruded with plastic cards, and 30 seeds were sown in each gap in evenly spaced positions, which was then covered with corresponding rhizosphere soil. For the seeds buried at 0 cm, seeds were placed on the surface of the soil and gently pressed to the soil by hand. Tap water was supplied daily to keep the soil moist.

During the experiment, seedling emergence was recorded daily. A seed was coded as "germinated" if the new shoots exceeded 1 cm in length. The experiment was terminated 30 days after sowing.

### 2.5. Influence of Cropping Systems on Reproductive Characteristics of P. minor

To test how different crops would impact the reproductive [25] and physiological characteristics [26] of this weed, we carried out a field experiment at the experimental farm at Yunnan Academy of Agricultural Sciences in Kunming City, China, from October 2017 to May 2018. Treatments comprised the three test crops (wheat, faba bean and rapeseed) and the weed (*P. minor*) in pure and mixed cultures (four proportions), utilizing a de Wit replacement series method [27].

The seeds of wheat, faba bean, rapeseed and *P. minor* were planted in plastic nursery seedling trays ($54 \times 28 \times 5$ cm), which had 50 holes in the greenhouse on 8 October 2017. After 30 days, similar-sized seedlings of the four plant species were transplanted. Four ratios of crops and *P. minor* plants were established (2:1, 1:1, 1:2, 0:3) while maintaining a constant planting density of 36 plants $m^{-2}$ ($0.20 \times 0.20$ m spacing). All plots were arranged in a complete randomized design with 4 replicates in $4\ m^2$ plots ($2 \times 2$ m). All crops and *P. minor* plants were distributed evenly within the plot. During the experiment, the weeds within the plot were removed manually, except for *P. minor*. The experimental plots were fertilized with 750 kg $ha^{-1}$ calcium superphosphate and 150 kg $ha^{-1}$ urea before planting and 75 kg $ha^{-1}$ urea at 60 days after transplanting. During peak flowering times, twenty main spikes of *P. minor* in each plot were randomly sampled and wrapped using net bags. After 120 days of transplantation, the number and dry weight of *P. minor* seeds were measured.

We harvested the plants at 120 d following transplantation. Twenty *P. minor* plants were selected randomly and harvested within the middle region of each plot. First, we counted the number of spikes, 40 fully expanded leaves (flag leaf and the top second leaf) were selected, and leaf area (LA) was determined using a portable laser leaf area instrument (CID-203, Li-Cor) [26]. The shoots, spikes and roots of twenty *P. minor* plants were sampled and placed in separate paper bags. Harvested plant parts were oven-dried at 80 °C for 48 h and then weighed. Specific leaf area (SLA) ($cm^2\ g^{-1}$) was calculated as area per unit dry mass. The ratio of leaf biomass to total biomass yielded the leaf weight ratio (LWR). Leaf area ratio (LAR) was calculated as the ratio of leaf area to plant weight.

During peak flowering times, fully expanded sun leaves (flag leaf and the top second leaf) of each species were selected for gas exchange measurements [27]. At the same time, chlorophyll fluorescence and gas exchange parameters were determined using the Li-6400XT portable gas exchange system (LI-COR Inc., Lincoln, NB, USA). Photosynthetic active radiation (PAR) was obtained by using a quartz halogen light unit coupled to a leaf chamber. During measurements, the relative air humidity 60%–70%, leaf temperature $25 \pm 1$ °C, and vapor pressure deficit (VPD) $2.0 \pm 0.5$ k Pa. Net photosynthetic rate (Pn), intercellular $CO_2$ concentration (Ci), stomatal conductance (Gs), transpiration rate (Tr) and water use efficiency (WUE) were determined [28,29].

### 2.6. Statistical Analysis

The germination data were recorded until the 15th day, we calculated the germination ratio and germination index of *P. minor* using the following parameters [22]:

(1)　Germination rate (%) = (Total number of germinated seeds/total seed number) $\times$ 100.

(2)　Germination index (*GI*) = $\Sigma$Gt/Dt (Gt is the germination number at the *t* day, Dt is the corresponding number of germination days).

Reproductive characteristics of *P. minor* were calculated using the following parameters [25]:

(1)  Reproductive allocation (g·g$^{-1}$) = spike biomass/total biomass of each plant.
(2)  Reproductive investment (g·g$^{-1}$) = seed biomass/total biomass of each plant.
(3)  Reproductive index (g·g$^{-1}$) = seed biomass/spike biomass of each plant.

All growth variables (seed germination number, germination index, leaf area, spike number, seed number spike biomass, seed biomass and total biomass) and physiological variables (Pn, Gs, Ci, Tr and WUE) of *P. minor* were analyzed by analysis of variance (One-way ANOVA) using the SPSS 23.0 software package (SPSS, Inc., Chicago, IL, USA). I Data were checked for homogeneity of Variance. Treatment means were separated with Tukey's HSD and Post Hoc Multiple Comparisons, at the 0.05 significance level.

## 3. Results

### 3.1. Seed Dormancy Characteristics of P. minor under Different Crop Types and Tillage Methods

The results showed that the seed germination rates of *P. minor* varied significantly under different crop types (Table 1). After 30 days of seed dispersal, the germination rates (less than 10%) of *P. minor* in the rice field were significantly lower than in the maize field and dry, bare soil conditions. Under the maize field and dry, bare soil conditions, the germination rates of *P. minor* increased significantly ($p < 0.05$) every month from seed dispersal to 120th day, over the length of the treatment, however, in the rice field (seed kept continuous submerged in water), the germination rates were less than 10% and leveled off rapidly over the season.

**Table 1.** Effects of summer crops on seed dormancy of *P. minor*.

| Treatment | Germination Rate under Different Buried Period (%) | | | | | |
|---|---|---|---|---|---|---|
| | 0 d | 30 d | 60 d | 90 d | 120 d | 150 d |
| A | 2.7 ± 0.3e | 20.0 ± 1.2d | 60.3 ± 2.2c | 65.9 ± 2.4b | 71.1 ± 2.8a | 74.4 ± 2.4a |
| B | 2.7 ± 0.3e | 11.9 ± 0.9d | 35.1 ± 1.4c | 73.3 ± 2.6b | 81.9 ± 2.2a | 75.3 ± 2.6a |
| C | 2.7 ± 0.3a | 2.7 ± 0.3a | 4.3 ± 0.4a | 4.7 ± 0.2a | 5.1 ± 0.2a | 5.1 ± 0.3a |

Data are expressed as mean ± standard error. The different letters within the same row mean significant differences at $p < 0.05$. (A) Seed buried in dry soil (never watered); (B) Seeds buried in the maize field (watered once every 5 days); (C) Seeds buried in rice field (continuously submerged in water).

### 3.2. Seed Germination Characteristics of P. minor under Different Crop Types and Tillage Methods

Crops of wheat, faba bean and rapeseed had no influence on the germination rate of *P. minor* seed, but significantly influenced ($p < 0.05$) its germination index, which was lowest in rapeseed treatment for the three upper burial depths (Table 2). The germination rate and germination index of *P. minor* seed were significantly affected by burial depth. Increasing soil depth significantly decreased the germination rate and germination index of this weed, but a few seeds germinated even at a burial depth of 12 cm.

**Table 2.** Effects of crops and burial depth on germination rate and germination index of *P. minor*.

| Items | Accompanying Plant | Burial Depth (cm) | | | | |
|---|---|---|---|---|---|---|
| | | 0.0 | 3.0 | 6.0 | 9.0 | 12.0 |
| Germination Rate (%) | Wheat | 96.0 ± 0.7a | 82.0 ± 2.4b | 67.3 ± 4.3c | 29.0 ± 3.5d | 4.5 ± 1.6e |
| | Faba bean | 95.3 ± 1.2a | 84.0 ± 2.0b | 67.0 ± 4.4c | 30.0 ± 2.8d | 5.3 ± 0.8e |
| | Rapeseed | 94.3 ± 1.1a | 84.0 ± 0.9b | 60.3 ± 3.5c | 31.0 ± 2.4d | 5.5 ± 0.5e |
| | Control | 96.3 ± 0.9a | 84.8 ± 0.5b | 69.5 ± 1.6c | 31.35 ± 1.2d | 5.8 ± 0.9e |
| Germination Index | Wheat | 19.1 ± 0.2a | 11.6 ± 0.5b | 6.9 ± 0.4c | 2.7 ± 0.3d | 0.3 ± 0.1e |
| | Faba bean | 18.6 ± 0.5a | 11.7 ± 0.4b | 6.8 ± 0.4c | 2.8 ± 0.3d | 0.4 ± 0.0e |
| | Rapeseed | 15.5 ± 0.4a | 10.0 ± 0.1b | 5.8 ± 0.4c | 2.7 ± 0.3d | 0.3 ± 0.0e |
| | Control | 19.8 ± 0.5a | 11.5 ± 0.4b | 7.3 ± 0.1c | 3.0 ± 0.1d | 0.4 ± 0.1e |

Data are expressed as mean ± standard error. The different letters within the same row mean significant differences at $p < 0.05$.

### 3.3. Effects of Crop Types and Tillage Methods on Reproductive Characteristics of P. minor

The growth and reproduction of *P. minor* varied greatly depending on crop environment (Table 3). In mixed culture, the total plant biomass, biomass and number of spike and seed number of *P. minor* were significantly ($p < 0.05$) suppressed with increasing proportions of wheat and rapeseed. However, faba bean was less suppressive than the other crops, with the spike and seed number of *P. minor* actually higher than in monoculture at ratios of faba bean to *P. minor* of 1:1 and 1:2.

Crop type and its relative abundance also elicited significant effects on reproductive allocation and reproductive investment of *P. minor* (Table 3). Under the same conditions, the reproductive allocation and reproductive investment of this weed were greatest when grown with faba bean and were the smallest when grown with rapeseed. With proportional increases in wheat and rapeseed, reproductive allocation and reproductive investment of *P. minor* became significantly lower ($p < 0.05$). However, in a mixed culture with faba bean, for the ratio of faba bean to *P. minor* of 1:1 and 1:2, the reproductive allocation exhibited no significant changes compared with the monoculture.

**Table 3.** Reproductive characteristics of *P. minor* growing as a monoculture and mixed culture conditions at different crop:weed ratios.

| Variables | | Ratios (Crops: *P. minor*) | | | |
|---|---|---|---|---|---|
| | | 2:1 | 1:1 | 1:2 | 0:3 |
| Total biomass (g) | Wheat | 4.7 ± 0.1d | 5.7 ± 0.1c | 6.1 ± 0.1b | 6.5 ± 0.1a |
| | Faba bean | 6.0 ± 0.1b | 6.9 ± 0.2a | 6.8 ± 0.1a | 6.5 ± 0.1a |
| | Rapeseed | 3.2 ± 0.1d | 4.0 ± 0.1c | 5.3 ± 0.1b | 6.5 ± 0.1a |
| Spike number | Wheat | 6.9 ± 0.2c | 7.4 ± 0.2c | 9.8 ± 0.3b | 12.3 ± 0.2a |
| | Faba bean | 7.7 ± 0.2d | 10.6 ± 0.3b | 9.9 ± 0.1c | 12.3 ± 0.2a |
| | Rapeseed | 3.5 ± 0.1c | 4.0 ± 0.1c | 6.5 ± 0.3b | 12.3 ± 0.2a |
| Seed number | Wheat | 547.8 ± 31.0d | 805.8 ± 35.7c | 982.5 ± 29.7b | 1167.8 ± 44.4a |
| | Faba bean | 777.2 ± 25.7b | 1190.2 ± 15.1a | 1181.0 ± 67.5a | 1167.8 ± 44.4a |
| | Rapeseed | 283.6 ± 17.4c | 307.4 ± 25.7c | 597.8 ± 58.2b | 1167.8 ± 44.4a |
| Spike biomass (g) | Wheat | 1.28 ± 0.02d | 1.83 ± 0.03c | 2.08 ± 0.04b | 2.57 ± 0.03a |
| | Faba bean | 2.07 ± 0.04c | 2.76 ± 0.06a | 2.73 ± 0.03a | 2.57 ± 0.03b |
| | Rapeseed | 0.79 ± 0.02c | 1.01 ± 0.04c | 1.41 ± 0.04b | 2.57 ± 0.03a |
| Seed biomass (1000-grain weight/g) | Wheat | 1.44 ± 0.01b | 1.47 ± 0.02b | 1.49 ± 0.02a | 1.52 ± 0.02a |
| | Faba bean | 1.50 ± 0.01a | 1.53 ± 0.02a | 1.52 ± 0.03a | 1.52 ± 0.02a |
| | Rapeseed | 1.36 ± 0.03b | 1.38 ± 0.04b | 1.41 ± 0.03b | 1.52 ± 0.02a |
| Reproductive allocation (g g$^{-1}$) | Wheat | 0.27 ± 0.01c | 0.32 ± 0.01b | 0.34 ± 0.01b | 0.40 ±0.01a |
| | Faba bean | 0.34 ± 0.00b | 0.40 ± 0.01a | 0.40 ± 0.01a | 0.40 ± 0.01a |
| | Rapeseed | 0.25 ± 0.01b | 0.25 ± 0.01b | 0.26 ± 0.01b | 0.40 ± 0.01a |
| Reproductive investment (g g$^{-1}$) | Wheat | 0.17 ± 0.01d | 0.21 ± 0.01c | 0.24 ± 0.01b | 0.27 ± 0.01a |
| | Faba bean | 0.19 ± 0.01b | 0.27 ± 0.01a | 0.26 ± 0.02a | 0.27 ± 0.01a |
| | Rapeseed | 0.12 ± 0.01c | 0.11 ± 0.01c | 0.16 ± 0.02b | 0.27 ± 0.01a |
| Reproductive index (g g$^{-1}$) | Wheat | 0.62 ± 0.03a | 0.65 ± 0.02a | 0.71 ± 0.04a | 0.69 ± 0.03a |
| | Faba bean | 0.56 ± 0.03b | 0.66 ± 0.01a | 0.66 ± 0.04a | 0.69 ± 0.03a |
| | Rapeseed | 0.49 ± 0.03b | 0.43 ± 0.04b | 0.59 ± 0.05a | 0.69 ± 0.03a |

Data are expressed as mean ± standard error. The different letters within the same row mean significant differences at $p < 0.05$.

### 3.4. Effects of Crop Types and Tillage Methods on Physiological Traits of P. minor in Reproductive Stages

Leaf area (LA), specific leaf area (SLA) and specific leaf weight (SLW) of *P. minor* were significantly affected by crop type and relative abundance in mixed culture (Table 4). In mixed culture, LA and SLW of *P. minor* were significantly ($p < 0.05$) suppressed with decreasing proportions of wheat and rapeseed, but the SLA of *P. minor* in mixed culture was greater than that in monoculture. The LA and SLW of *P. minor* in mixed tended to be highest in association with faba bean. The leaf area and specific leaf

weight of *P. minor* were lowest in association with rapeseed, but SLA highest for *P. minor* in association with rapeseed.

**Table 4.** Leaf morphology and physiological traits of *P. minor* growing as in monoculture and in mixed culture in the reproductive stage measured 120 d after planting.

| Variables | | Ratios (Crops: *P. minor*) | | | |
|---|---|---|---|---|---|
| | | 2:1 | 1:1 | 1:2 | 0:3 |
| Leaf area (cm$^2$) | Wheat | 7.0 ± 0.2c | 7.3 ± 0.3c | 8.4 ± 0.4b | 9.3 ± 0.2a |
| | Faba bean | 9.2 ± 0.1a | 9.4 ± 0.3a | 9.3 ± 0.3a | 9.3 ± 0.1a |
| | Rapeseed | 4.9 ± 0.2d | 6.7 ± 0.2c | 7.7 ± 0.2b | 9.3 ± 0.1a |
| Specific leaf area (SLA) (cm$^2$ g$^{-1}$) | Wheat | 344.3 ± 8.5a | 332.1 ± 13.7a | 298.9 ± 7.6b | 291.1 ± 3.8b |
| | Faba bean | 315.5 ± 3.5a | 283.1 ± 4.9b | 294.2 ± 11.9b | 291.1 ± 3.8b |
| | Rapeseed | 370.4 ± 12.1a | 344.8 ± 10.8b | 322.6 ± 9.1c | 291. ± 3.8d |
| Specific leaf weight (SLW) (mg$^1$cm$^{-2}$) | Wheat | 3.2 ± 0.1c | 3.4 ± 0.1bc | 3.7 ± 0.2b | 4.2 ± 0.1a |
| | Faba bean | 4.4 ± 0.1b | 5.2 ± 0.2a | 5.4 ± 0.2 | 4.2 ± 0.1b |
| | Rapeseed | 2.7 ± 0.1c | 2.9 ± 0.1c | 3.1 ± 0.1b | 4.2 ± 0.1a |
| Net photosynthetic rate (Pn) (umol CO$_2$ m$^{-2}$ s$^{-1}$) | Wheat | 15.1 ± 0.3c | 16.1 ± 0.3b | 16.5 ± 0.2b | 18.7 ± 0.4a |
| | Faba bean | 17.9 ± 0.2b | 18.1 ± 0.1b | 18.8 ± 0.2a | 18.7 ± 0.4ab |
| | Rapeseed | 14.4 ± 0.3c | 14.8 ± 0.2c | 15.7 ± 0.2b | 18.7 ± 0.4a |
| Stomatal conductance, (Gs) (mol H$_2$O m$^{-2}$ s$^{-1}$) | Wheat | 0.32 ± 0.01b | 0.34 ± 0.02ab | 0.35 ± 0.01ab | 0.37 ± 0.01a |
| | Faba bean | 0.34 ± 0.01a | 0.36 ± 0.01a | 0.37 ± 0.02a | 0.37 ± 0.01a |
| | Rapeseed | 0.29 ± 0.01c | 0.29 ± 0.02c | 0.33 ± 0.01b | 0.37 ± 0.01a |
| Intercellular CO$_2$ conductance (Ci) (umol mol$^{-1}$) | Wheat | 294.3 ± 4.3a | 291.6 ± 2.8ab | 288.4 ± 2.7ab | 283.5 ± 2.2b |
| | Faba bean | 286.0 ± 2.4a | 284.2 ± 3.8a | 280.4 ± 39.4a | 283.5 ± 2.2a |
| | Rapeseed | 289.3 ± 3.7a | 288.5 ± 3.7a | 292.3 ± 1.7a | 283.5 ± 2.2a |
| Transpiration rate (Tr) (mmol H$_2$O m$^{-2}$ s$^{-1}$) | Wheat | 4.0 ± 0.0c | 4.2 ± 0.1bc | 4.3 ± 0.0b | 4.9 ± 0.1a |
| | Faba bean | 4.7 ± 0.1b | 4.9 ± 0.1ab | 5.0 ± 0.1a | 4.9 ± 0.1ab |
| | Rapeseed | 3.9 ± 0.1c | 3.8 ± 0.1c | 4.2 ± 0.1b | 4.9 ± 0.1a |
| Water use efficiency (WUE) (umol CO$_2$ mmol H$_2$O $^{-1}$) | Wheat | 3.8 ± 0.1a | 3.9 ± 0.1a | 3.9 ± 0.1a | 3.8 ± 0.1a |
| | Faba bean | 3.8 ± 0.1a | 3.74 ± 0.1a | 3.7 ± 0.1a | 3.8 ± 0.1a |
| | Rapeseed | 3.7 ± 0.1a | 3.87 ± 0.1a | 3.7 ± 0.1a | 3.8 ± 0.1a |

Data are expressed as mean ± standard error. The different letters within same row mean significant differences at $p < 0.05$.

Crop types and relative abundance also significantly influenced the physiological characteristic of *P. minor* (Table 4). When *P. minor* was grown with wheat or rapeseed, the net photosynthetic rate (Pn), stomatal conductance (Gs) and transpiration rate (Tr) decreased significantly compared to *P. minor* grown in monoculture (but note that Gs was only higher for monoculture compared to wheat in the 2:1 mix). However, there were no significant differences in Gs for *P. minor* grown with faba bean, although the Tr was largest for a 1:2 ratio crop to *P. minor*. The intercellular CO$_2$ concentration (C$_I$) varied little for *P. minor* in mixed culture with faba bean and rapeseed, but at a 2:1 ratio of wheat to *P. minor* C$_i$ was significantly higher than in monoculture. There were no significant differences in water use efficiency (WUE) between mixed culture and monoculture. To sum up the results: the Pn, Gs and Tr of *P. minor* in association with faba bean and wheat were the greatest, while the Pn, Gs and Tr for *P. minor* in association with rapeseed were the smallest.

## 4. Discussion

Control of invasive weeds utilizing cropping systems is an important current research topic [13,30,31], but specific information on managing *P. minor* via such an approach is limited. For annual weeds like *P. minor*, understanding the seed dormancy, germination reproductive characteristics under different cropping systems is critical for designing appropriate systems that provide ecological advantages allowing pesticide use to be reduced [32–34]. Among the cropping systems tested in our dormancy experiments, we found crop types significantly affected the seed

dormancy of this weed, as seeds can remain dormant for a long time in paddy fields (less than 10% germination by the 150th day). In our seed germination experiment, we found that increasing soil depth significantly decreased the germination rate and germination index of *P. minor*. Our field experiment showed rapeseed significantly reduced the total biomass, spike biomass, spike number and seed number of *P. minor*. Together, these results suggested that cropping systems could be designed to regulate seed dormancy, inhibit seed germination and reduced the soil seed banks of *P. minor*.

Weed seed dormancy is a critical issue for weed control and varies greatly under different environmental conditions [13]. Previous studies found that seed dormancy of *P. minor* could be broken via specific conditions involving sunlight, temperature, or soil moisture [15,18]. Our research showed that seed dormancy of *P. minor* was affected by different cropping systems, as *P. minor* exhibited higher germination rates in maize fields and bare, dry soil and exhibited extremely low germination rates in submerged paddy fields (<10%). The main crops grown in Yunnan during the dormancy period of *P. minor* are rice and maize. Our results suggested that before sowing winter crops, if the previous crop was maize or fallow, the control of this weed could be timed to coincide with *P. minor* emergence. However, if the previous crop was rice, the water level could be kept elevated as long as possible before sowing winter crops, so as to delay the emergence of this weed, and put it at a disadvantage. Thus, strategically rotating other systems such as paddy rice with rapeseed could reduce *P. minor* germination and help deplete its seed bank.

Weed seed germination may be greatly affected by soil microclimate conditions, such as temperature and soil moisture, as influenced by crop type. Some studies found that some crops, such as pearl millet, black gram, green gram, sorghum, soybean and sunflower could delay and inhibit seed germination of *P. minor* [32–34]. However, our study showed three crops (wheat, faba bean and rapeseed) had no effect on the seed germination percentage of *P. minor*, but that its germination time was delayed significantly by rapeseed. Some crops are confirmed to have inhibitory effects on the germination of *P. minor* and can be further explored as natural alternatives to chemical herbicides [35,36]. Our previous research showed that rapeseed had strong competitive ability against *P. minor* while the current research suggests that rapeseed may gain a competitive advantage over *P. minor* by delaying its germination [23]. In addition, we did also show that increasing soil burial depth significantly reduced seed germination of *P. minor* implying that agricultural measures such as deep-tillage could reduce *P. minor* populations.

Our research also showed that rapeseed could significantly reduce the spike biomass, spike number and seed number of *P. minor*, while low-density faba bean could actually promote these growth characteristics. Previous studies showed that *P. minor* had high reproductive capacity and robust ecological adaptability [5]. We have previously studied the competitive effects between native crops (wheat, faba bean and rapeseed) and *P. minor*, and these studies showed that rapeseed was the strongest while faba bean was the weakest under the same conditions [23]. Therefore, utilizing interspecific competition the seed number of *P. minor* may be reduced by planting highly competitive species, thus overcoming the inherent adaptability of this invasive species, as normally realized via high levels of fecundity. Moreover, our research also showed the reproductive allocation, reproductive investment and reproductive index of *P. minor* in rapeseed fields were the lowest while *P. minor* exhibited the highest reproductive potential in association with faba bean. Many invasive alien species have strong ecological adaptability, allowing them to allocate resources strategically to manage the inevitable trade-off between growth and reproduction [37].

Functioning leaf area can reflect a plant's strategy for maximizing the capture and utilization of environmental resources [38]. Thus, leaf area is generally useful in assessing the growth condition and solar energy utilization efficiency of plants [39]. Greater specific leaf area (leaf area per unit leaf mass) may increase carbon assimilation due to more leaf area produced for a given investment in biomass [40]. Our study showed that in all mixed culture treatments, the leaf area and specific leaf weight of *P. minor* were lowest in rapeseed fields and highest in faba bean fields, indicating that rapeseed was the most competitive crop. Higher rates of photosynthesis connected to higher leaf areas

can lead to increased growth rates, biomass accumulation and overall production. Higher carbon gain and growth may enable many invasive species to readily outcompete slower-growing species by facilitating colonization or resource acquisition [40]. The net photosynthetic rate, stomatal conductance and transpiration rate of *P. minor* also decreased significantly while grown with wheat or rapeseed. We found the Pn of *P. minor* was significantly higher in monoculture than in mixed culture with rapeseed and wheat, demonstrating that interspecific competition reduced its rate of photosynthesis. Thus, it is clear that impacts of rapeseed on the structural and physiological traits of *P. minor* were generally more pronounced than for wheat and faba bean, suggesting that growing rapeseed crops may be more effective at inhibiting *P. minor* than wheat or faba bean.

Our results also have significant implications for ecological control theory and more general applications to management of invasive weeds in agro-ecosystems. Ecological control of weeds in farmland is still a worldwide challenge because of human demand for food and high economic returns. Some invasive weeds have highly persistent soil seed banks in invaded habitats. Our research showed that cropping system could affect the seed dormancy of invasive weeds, and the design of cropping system could regulate the temporal and spatial distribution of invasive weed populations, which is conducive to the ecological control of weeds. In addition, our research also showed that planting strong competitive crops could reduce the harmfulness of invasive weeds and minimize the use of herbicides. Some researchers also argue that certain cropping systems may be more successful than others in resisting invasion and provide ecological control [41]. Compared to manual uprooting and chemical control, ecological control clearly has the potential to provide a more sustainable management option for growers, as shown in the present study and other related studies [42,43]. Our research revealed that rapeseed can significantly restrain growth, reduce seed number and increase germination time of *P. minor*. In view of our findings on the impact of microsite conditions on the germination and growth of *P. minor*, ecological control measures should be designed to involve both appropriate crop rotations and tillage management, in order to manage the ecosystem effectively.

## 5. Conclusions

This study demonstrated the roles of selected crop types and seed and seedling microclimate conditions in potentially regulating seed dormancy, inhibiting seed germination and the reproductive ability of *P. minor*, thus reducing establishment and spread of *P. minor* populations. Our findings suggest optimal cropping systems design could involve planting rapeseed in conjunction with deep plowing and planting rice (continuous submergence underwater) in summer. Such a system could reduce the field populations and seed bank of *P. minor*, thus providing a sustainable and environmentally friendly means of suppressing *P. minor*.

**Author Contributions:** L.D. and F.Z. conceived and designed the experiments; G.X., S.S., S.Y., Y.Z., G.J. and Y.G. performed the experiments; G.X., D.R.C. and J.L. analyzed the data and wrote drafts; S.S. and D.R.C. edited the manuscript for style; L.D. and F.Z. commented on manuscript. All authors read and approved the final manuscript.

**Funding:** This research was supported by the Applied Basic Research Foundation of Yunnan Province (2017FB055), the Middle-aged and Young Academic Leader training foundation of Yunnan Province (2018HB054), the Ten Thousand Talent Program (Young Top-notch Talent) of Yunnan Province (2019–2023), the Applied Basic Research Foundation of Yunnan Province (2017FB049), the State Scholarship Fund of China Scholarship Council (201808530029), the Program for the Innovative Research Team of Yunnan Province (2020–2022) and the Key Research and Development Program of Yunnan Province (2019IB007).

**Acknowledgments:** We wish to thank Yuhua Zhang from the Agricultural Environment and Resource Research Institute, Yunnan Academy of Agricultural Sciences for her great field support.

**Conflicts of Interest:** The authors declare no conflict of interest. The founding sponsors had no role in the design of the study; in the collection, analyses, or interpretation of data; in the writing of the manuscript and in the decision to publish the results.

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
