# Peer review of "Designing Cropping Systems to Improve the Management of the Invasive Weed Phalaris minor Retz."

_agronomy, doi:10.3390/agronomy9120809_

Round 1

Reviewer 1 Report

The topic of this manuscript is very interesting.  Authors, examined the effects of different cropping systems on  seed germination, growth and physiological parameters of Phalaris minor. I consider that the manuscript contains information's that deserve to be published after minor revision.

I provide below a few suggestions that, if the authors decide to implement into the paper, the paper will improved.

Comments

Title: The title of the article should revised. e.g., Designing cropping systems to improve the management of the invasive weed Phalaris minor Retz.

Abstract: In general the abstract is well written.

Line 28: Replace earhead with panicle.

Line: Replace while with when.

Introduction: The introduction section is well written.

Line 44: canarygrass(Poaceae) should be corrected as canarygrass (Poaceae)

Material and methods:

Line 120: Authors should explain why the maize crops was planted at 20 x 20 cm since maize is planted at 75 x 15-20 cm. The dates of sowing should be added in each experiment.

Results

This section is well written.

Corrections

In table 1 the germination rate at 0 day is not needed. I can’t understand why seeds of P. minor was germinated at 0 days.

In table 3 and 4 in the last treatment 0:3 and for each variable is not needed to repeat the number for wheat, faba bean and rapaseed treatement. It is difficult in agronomic experiments to have the same mean value for a growth parameter from the same number of replications and the same treatment. The mean value for plant biomass for wheat, faba bean and rapeseed are from the same replications or from different;

Discussion: This section should be revised, more references about the effects of crops density on growth and physiology of P. minor or other winter grass weeds should be added. Authors should explain why in rice treatment the seed germination is reduced.

References

The references should be corrected following the instructions for authors. For example in the references 6,7 the journal names should be correctly abbreviated.

Author Response

Title: The title of the article should revised. e.g., Designing cropping systems to improve the management of the invasive weed Phalaris minor Retz.

DONE

Abstract: In general the abstract is well written.

Line 28: Replace earhead with panicle.

CHANGED TO “spike”

Line: Replace while with when.

DONE

Introduction: The introduction section is well written.

Line 44: canarygrass(Poaceae) should be corrected as canarygrass (Poaceae)

Material and methods:

Line 120: Authors should explain why the maize crops was planted at 20 x 20 cm since maize is planted at 75 x 15-20 cm. The dates of sowing should be added in each experiment.

YES, THE MAIZE USUALLY IS PLANTED AT 75 15-20 CM, BUT WE WANT TO MAKE THE RICE AND MAIZE HAVE SAME DENSITY DURING THE EPXERIMENTS. THE DATES OF SOWING WAS ADDED

Results

This section is well written.

Corrections

In table 1 the germination rate at 0 day is not needed. I can’t understand why seeds of P. minor was germinated at 0 days.

0 DAY IS BURIED DAYS. SO WE CHANGED THE ‘Germination rate’ TO ‘Germination rate under different buried period’ IN TABLE 1

In table 3 and 4 in the last treatment 0:3 and for each variable is not needed to repeat the number for wheat, faba bean and rapaseed treatement. It is difficult in agronomic experiments to have the same mean value for a growth parameter from the same number of replications and the same treatment. The mean value for plant biomass for wheat, faba bean and rapeseed are from the same replications or from different;

THE MEAN VALUES IN THE LAST TREATMENT 0:3 ARE CORRECT AND ARE FROM THE SAME REPLICATIONS. THE REASON THEY ARE CONSISTENT IS THAT THERE ARE NO CROPS IN THESE TREATMENTS.

Discussion: This section should be revised, more references about the effects of crops density on growth and physiology of P. minor or other winter grass weeds should be added. Authors should explain why in rice treatment the seed germination is reduced.

HERE ARE SOME REFERENCES RELATING TO P. MINOR WE ADDED:

Abbas, T.; Nadeem, M.A.; Tanveer, A.; Ali, H.H.; Safdar, M.; Zohaib, A.; Farooq, N. Exploring the herbicidal and mormetic potential of allelopathic crops against fenoxaprop-resistant Phalaris minorPlanta Daninha 2018, 36, e018176368. http://dx.doi.org/10.1590/s0100-83582018360100056.

El-Darier, S.M.; El-Kenany, E.T.; Abdellatif, A.A.; Hady, E.N.F.A. Allelopathic prospective of Retama raetam L. against the noxious weed Phalaris minor Retz. growing in Triticum aestivum L. fields. Rend. Fis. Acc. Lincei 2018, 29, 155-163. https://doi.org/10.1007/s12210-018-0675-x

Vivek; Naresh, R.K.; Tomar, S.K.; Kumar, S.; Mahajan, N.C.; Shivani. Weed and water management strategies on the adaptive capacity of rice-wheat system to alleviate weed and moisture stresses in conservation agriculture: a review. Int. J. Chem. Stud. 2019, 7, 1319-1334. http://www.chemijournal.com/archives/?year=2019&vol=7&issue=1&ArticleId=4896&si=false

Malik, R.K.; Kumar, V.; McDonald, A. Conservation agriculture-based resource-conserving practices and weed management in the rice-wheat cropping systems of the Indo-Gangetic Plains. Indian J. Weed Sci. 2018, 50, 218-222. http://dx.doi.org/10.5958/0974-8164.2018.00051.5

References

The references should be corrected following the instructions for authors. For example in the references 6,7 the journal names should be correctly abbreviated.

THE REFERENCES WERE CORRECTED

Reviewer 2 Report

The authors answered that have accepted the reviewer's suggestions, but then they didn't revised the manuscript, especially for M&M section and Statistical analysis. In fact, the experiments were not repeated and the statistical analysis was not revised as suggested.

Author Response

The authors answered that have accepted the reviewer's suggestions, but then they didn't revised the manuscript, especially for M&M section and Statistical analysis. In fact, the experiments were not repeated and the statistical analysis was not revised as suggested.

OUR STATISTICAL ANALYSIS IS APPROPRIATE FOR THIS KIND OF DATA.

Reviewer 3 Report

Please refer to the attached file

Author Response

There are still many language issues in the manuscript, therefore I have added corrections in the pdf (inserted in the end of this file). This have shifted the text a bit, therefore the line numbers of this document refers to the pdf with corrections.

THESE ISSUES HAVE ALL BEEN ADDRESSED AND ADDITIONAL EDITING HAS BEEN DONE TO IMPROVE THE WORDING THROUGHOUT

General comment: The manuscript is significantly improved, though I still find it hard to follow the discussion. The coherency of the written text decrease through the discussion. And there are repetitions. The fact that oilseed rape is considered beneficial for controlling P. minor is stated several times. 

Specific comments (additionally see the attached pdf):

20-22: this sentence does not make sense. Please, rephrase.

THE ORIGINAL SENTENCE “Here, we studied seed dormancy, germination characteristics and reproductive responses of P. minor to various cropping systems to show how ecological control this weed utilized cropping systems in China.” IS NOW CHANGED TO: “Here, we studied seed dormancy, germination characteristics and reproductive responses of P. minor to various cropping systems to show how cropping systems could be better designed to control P. minor in China.”

27: and here the oilseed rape differed from the other crops

SENTENCE CHANGED TO: “Total biomass, spike biomass, spike number and seed number of P. minor were significantly reduced (P<0.05) with increasing proportions of the three crops (wheat, faba bean and rapeseed), with rapeseed having the strongest inhibition effects among the three crops.”

34: the use of “moreover” indicate that this is more important than the above mentioned results, is this the case in your opinion?

MOREOVER DOES NOT NECESSARILY INFER THERE IS SUCH A HIERARCHY; AT ANY RATE, WE ARE KEEPING THE WORDING THE SAME

70: this sentence need rephrasing. What is wrapped in foil?

CHANGED TO: “For example, the germination of P. minor seed was higher when the seed was exposed to white light after dispersal compared to germination in the dark (wrapped in aluminium foil) [18].”

82-83 and l. 85-86: these two sentences are repetitive, they state the same thing. Delete one.

THE FIRST ONE WAS DELETED.

88-89: this is not really a hypothesis. A hypothesis include what you expect. E.g. “We wanted to test the hypothesis that changing cropping system can contribute to the control of P. minor.”

SENTENCE CHANGED TO: “Specifically, our objective was to test the hypothesis that cropping systems could be designed to improve the control of the invasive weed P. minor in agroecosystems.”

130: why not maize and rice as in the dormancy study? Could you add a justification for this difference in choice of crops? I did not think of this, when I read the manuscript the first time.

GROWING SYSTEMS IN YUNNAN OCCUR IN TWO SEASONS (SUMMER AND WINTER, AS EXPLAINED AT THE END OF SECTION 2.2), IN WHICH MAIZE AND RICE ARE THE MAIN SUMMER CROPS, SO THE VARIETY OF CROPS WAS NECESSARY BECAUSE OF THE TIMING OF THE RESEARCH

167: you write in your response to my previous comments that your focus is the P. minor reproduction characteristics and therefore you only show the data on P. minor, not the crop species. But then why sample the crop species? Or if you want to use this data for something else, then why write that you sampled the crop data? Just leave that out of this manuscript. It is confusing to get the information that you sample the crop plants and then the data is not there.

THE SENTENCE WAS ALTERED TO REFER ONLY TO P. minor

Table 1: As written in the first round of review, it is standard to have one more decimal on the standard error value than on the measured value. I would suggest to have one decimal on the germination rate and index and two on the standard error.

DONE.

Table 2: same comments as for table 1

DONE

Table 3: I would consider the number of decimals or digits. E.g. Total biomass: is it really important whether the number is 4.74 or just 4.7? The more decimals, the more time the reader needs to spend reading the numbers. And for seed number: I would skip the decimals and just give the rounded whole number. But it is up to you, how precise you think the numbers needs to be.

DONE

233-234: as suggested in the previous review, I think you use decrease instead of increase (this also appears to fit the text in the abstract). As far as I can see, the values of total biomass, biomass and number of earhead is suppressed with increasing proportions of wheat and rapeseed?

CHANGED TO “with increasing proportions”

234-235: rephrase this sentence. I think you referring to the fact that faba bean is less suppressive than wheat and rapeseed and in fact higher than in monoculture, but it is not absolutely clear.

SENTENCE MODIFIED (USING A NEW SENTENCE) TO: “However, faba bean was less suppressive than the other crops,  with the spike and seed number of P. minor actually higher than in monoculture at ratios of faba bean to P. minor of 1:1 and 1:2.”

Table 4: I still don’t understand why you have the abbreviations in brackets after Specific leaf area and Specific leaf weight and not leaf area? I also think it would be nice to have the full text for Pn, Gs (it still says Cs in the table –correct), Ci, Tr and WUE in the table. The reader will not have to the text to find out that Pn is Net photosynthetic rate etc. There is still no explanation for WUE in 2.6. SLW is dry weight per unit of projected area add the unit in the row name.

THESE CHANGES WERE MADE. WUE AND SLW WERE EXPAINED IN 2.5.

251: The comments applies here related to decrease/increase as the values for LA and SLW are higher when lower proportions of crop plants are present. The opposite is the case for SLA, even though the differences are smaller. There are no lower case letters in row 1:2 for SLA, add the letters in table 4.

THE LETTERS WERE ADDED

258: Gs is only higher for monoculture compared to wheat in the 2:1 mix.

THIS IS NOW NOTED AT THE END OF THE SENTENCE, I.E. “(but note that Gs was only higher for monoculture compared to wheat in the 2:1 mix).”

259: rephrase, you write about the Gs in the beginning of the sentence and Tr in the end.

WE FAIL TO SEE WHAT THE REVIEWER MEANS – THIS SENTENCE DOES FLOW LOGICALLY, WITH Gs IS WRITTEN ABOUT IN THE BEGINNING AND Tr AT THE END

261: this should be Ci, not Gs.

CHANGED

264: the values are largest with faba bean and wheat, so I assume the written text is wrong –see my correction in text.

DONE

268: in the introduction you refer to some studies on biology, so not completely lacking, but limited?

WORDING CHANGED TO “limited”

273-276: are you aware of other literature that have similar or different findings? On this species or weeds in general? This could be a nice addition to the discussion.

DONE

287: what is wasteland? Do you mean set-a-side/ fallow?

CHANGE TO “fallow”

L 288: I do not know what you mean by centralized?

CHANGED TO: “Our results suggested that before sowing winter crops, if the previous crop was maize or fallow, the control of this weed could be timed to coincide with P. minor emergence.”

289: how do you maintain the soil moisture? Is that a special thing in rice paddy fields, like when you let the water out of the field?

SENTENCE CHANGED TO: “However, if the previous crop was rice, the water level could  be kept elevated as long as possible before sowing winter crops, so as to delay the emergence of this weed, and put it in a disadvantageous niche and reduce its harm.”

293: I would use the word “microclimate” instead of “micro-site conditions”, but does crop type belong to this?

SENTENCE CHANGED TO: “Weed seed germination may be greatly affected by soil microclimate conditions, such as temperature and soil moisture, as influenced by crop type.”

298: even though the statistical analysis show significant delay, what is the practical impact of the delay? How to interpret the germination index? What does the difference from 15.53 to 19.83 actually mean?

THE SEED GERMINATION RATE AND GERMINATION INDEX ARE WIDELY USED IN SEED STUDIES, SO NOT NEED INTERPRET TOO MUCH. LOWER GERMINATION INDEX MEANS LONGER DELAYED SEED GERMINATION WHICH MAY AFFECT THE LATER PLANT GROWTH AND COMPETITION.

299: rephrase, “gained favorable niche in interspecific competition” is not a good way to phrase it.

SENTENCE RE-WORDED TO: “Our current research may suggests that rapeseed may gain a competitive advantage over P. minor by delaying its germination.”

L 301: you already mentioned germination time/index above, do not repeat yourself.

REFERENCE TO GERMINATION INDEX OMITTED

303: you did not study the allopathic interactions directly.

REFERENCE TO ALLELOPATHY OMITTED

308: You miss a reference in this line.

REFERENCE ADDED

308-310: how does the present study differ from the previous? They sound very similar, but I do not have access to the reference [23]. This have to be new compared to the previous study to be published.

THE LINK WAS PROVIDED FOR THIS PAPER. THE PREVIOUS STUDY WAS FOCUSED ON THE COMPETITION RELATIONSHIP BETWEEN P. MINOR AND THREE CROPS, WHEREAS THE PRESENT STUDY WAS MORE IN DEPTH RESEARCH ON HOW THE COMPETITION WORKED (SEE ALSO THE SECOND LAST PARAGRAPH OF THE INTRODUCTION0

311: what does “those” relate to? SEE NEXT 313: “overcoming the inherent adaptability”? I’m not sure what you refer to here?

SENTENCE CHANGED TO: “Therefore, utilizing interspecific competition the seed number of P. minor may be reduced by planting highly competitive species, thus overcoming the inherent adaptability of this invasive species, as normally realized via high levels of fecundity.”

Between line 317 and 318: “All these” what do you refer to here?

SENTENCE OMITTED

resources [34].

318: “Leaf functional” – functional what?

SENTENCE CHANGED TO: “Functioning leaf area can reflect a plant’s strategy for maximizing the capture and utilization of environmental resources [34].”

319: major index?

SENTENCE CHANGED TO: “Thus, leaf area is generally useful in assessing the growth condition and solar energy utilization efficiency of plants [35].”

L 321-324: this sentence needs rephrasing. It seems to be correct in terms of the statement, but it is really difficult to read.

SENTENCE CHANGED TO: “Our study showed that in all mixed culture treatments, the leaf area and specific leaf weight of P. minor were lowest in rapeseed fields and highest in faba bean fields, indicating that rapeseed was the most competitive crop.”

348: this is repetitive and stated several times during the discussion, delete here.

OMITTED

352: I’m not sure what you mean by associated crop? Diversity in crop rotation?

CHANGED TO “appropriate crop rotations”

354: you have not demonstrated the influence of different tillage system, but you have observed that burial depth is important and therefore conclude that ploughing is a control option.

“tillage methods” CHANGED TO  “seed microclimate conditions”

356-366: you just repeating the content of the results and discussion, delete it. L. 366-269 is you conclusion.

DONE

Round 2

Reviewer 2 Report

The statistical analysis could be also appropriate but it is not the best statistical method that should be used. The use of regression analysis for the germination rate after different summer crops and burial depth are suggested. Furthermore, the experiments were not repeated and this is a limit for the results to support conclusions. 

Reviewer 3 Report

See comments in the file attached

This manuscript is a resubmission of an earlier submission. The following is a list of the peer review reports and author responses from that submission.

Round 1

Reviewer 1 Report

This manuscript addresses the effects of cultural management and interspecific competition; I am not sure that biotic resistance is the correct term.

Specific comments are made on the attached PDF.

The manuscript requires extensive editing for grammatical correctness.

Reviewer 2 Report

The topic of this manuscript is very interesting.  Authors, examined the effects of 1) soil environment on seed germination of P. minor, 2) rhizosphere soil of difference winter crops and sowing depth on P. minor germination and 3) the competitive ability of P. minor in three winter crops. I consider that the manuscript contains information's that deserve to be published but in the current form should be rejected. The authors failed to show the significance of their study.

I provide below a few suggestions that, if the authors decide to implement into the paper, the paper will improved.

Comments

Title: The title of the article should revised. In the current form it is not obvious which the aim of this manuscript.

Abstract: The abstract should be revised. Lines 17-18 should be deleted. The term farmland habitats is widely used in the abstract. In my opinion this term should not be used in the manuscript. The authors should clearly present which the aim of the current study and which experiments were conducted.

Introduction: The introduction section should be revised. The first paragraph (lines 41-50) should be deleted. The authors should add information about the effects of environment, soil conditions, soil depth etc. on seed germination or dormancy of Phalaris minor. Moreover, should be add information about the competitive ability and growth of P. minor in different winter crops.

Material and methods: This section is well written.

Minor comments:

Lines 155-166. The amount of fertilizers should be converted to Kg per ha.

Results

Section 3.2. is missing from the manuscript.

Table 1. The data of table 1 is not presented in the text.

Table 3. The abbreviate parameters should be described in the footnote of table.

Discussion: In this section the authors failed to show the significance of their study.

Lines 259-260: In these lines authors reported that control of invasive weeds with the biotic resistance is a hot research topic. The authors should analyze more this statement while it should be supported with several references. It is not clear how biotic resistance control the invasive weeds.

Lines 278-283: In these lines authors reported that seed dormancy of P. minor was affected by the field environment, while during the dormancy period of this weed rice and maize crops are the main crops in Yunnan.

I would like to underline that P. minor is a main weed in winter crops and usually at the summer period the seeds had high level of dormancy. So, usually this weed is not germinated in rice and maize crops. So, this control method that described by the authors in these lines is not exist.

More references about the effects of crops density on growth and physiology of P. minor or other winter grass weeds should be added.

References

The references should be corrected following the instructions for authors. For example in the reference 9 the journal name is not correct.

Reviewer 3 Report

General comments to the authors:

The manuscript treats an aspect that could be related to the aims of the journal, but there are some aspects that need to be implemented in this research paper, before to reconsider it for publication.

First. The methodologies

The experiments were carried out not in real conditions (i.e. cement pool, Petri dishes, plastic nursery) in order to determine the effects of biotic factors in native habitat on P. minor. This kind of approach is not representative of the ecological real conditions in the fields and/or agroecosystems. Furthermore, all experiments were not repeated, as needed in this kind of research.

For these reasons it is necessary to repeat the experiments in order to have a more robust dataset and so a good representativeness of results.

Second. The M&M session.

This section is very poor of information and not clearly wrote so it needs to be implemented in the following parts:

Lines 92-93, 96-97, 142-143, have to be moved to the introduction section; The study site description shows a very strange situation. The authors reported that in the experiment site there was not P. minor in the past two years, but this weed was the most serious invasive species in the wheat filed close to the field used as test. Please, could you explain this particular diffusion and distribution of P. minor?

The experiment “2.3 Influence of farmland habitats on seed dormancy of P. minor” has had the objective to evaluate the effects of environment on seed dormancy of P. minor. However, even if the dormancy can be affected by allelopathic effects of crops, the main effects are due to the environmental conditions, like burial depth of seeds, temperature and moisture of soil. For these reasons the experiment is not organized correctly to be able to evaluate the effect on seed dormancy of the crop systems (i.e. maize or rice), but simply able to evaluate the effect due to the soil conditions (dry, irrigated once every 5 day or completely submerged in water). The correct way to be able to separate the allelopathic effects to the soil conditions is to repeat the experiment, adding to the three treatments already used, other two treatments: i) seeds buried in soil only and no crop (water once every 5 days); ii) seeds buried in soil only and no crop (keep submerged in water). So, please, repeat the experiment according these suggestions. Furthermore, the authors not used any test to define the viability of un-germinated seeds (i.e. tetrazolium test). Without this test and so the data on the viability of seeds is not possible to know if un-germinated seeds are really dormant or died (for example, after a long period in water immersion conditions). So please, use this test in the next repeated experiment.   

Third. The Statistical analysis

The statistical analysis is not appropriate and correct. In fact, the Multiple Comparison Procedures (MCP) is fully justified in the case of experiment aimed at comparing a set of unrelated factor levels (e.g. different crop varieties), but it is inefficient in the case of a quantitative explanatory variable, like time in the first experiment 2.3 and buried depths in the second experiment 2.4. For these reasons, the Tukey test is not correct and it would be correctly used the regression analysis approach for germination rate vs times (see the first experiment) and for the germination rate and germination index vs different seed burial depths (see the second experiment).

Results and Discussion sections

These sections have to be re-wrote according to the data of the repeated experiment and the new statistical analysis.

Specific comments to the authors:

Line 46: replace invisibility with invasibility; Line 106: “water once every days”: how much water did you apply? and how did you apply water to the pool? Line 108: “4 cement pool (1,5x1,5 m)” and the depth?, Please add Lines 109-110: how many seeds were buried in each cement pool? Line 112: replace “No fertilizer or pesticide to use” with “no fertilizers or pesticides were used” Line 115: replace “sown” with “put” Line 116: how much water did you add in each Petri dish for germination test? Line 117: “seedling emergence”, why seedling emergence? In Petri dishes it should say “seed germination”. Please correct Line 117: replace “germination” with “germinated” Line 117: please add when you stopped the experiment. Line 123: please add the depth of plastic nursery Line 139: please, replace “germination” with “emerged” Line 164: “shoots, earhead and roots of twenty…..were sampled…..”. How did you sample the roots without damage them? Please, explain Line 177: “The data on germination was recorded until the fifteenth days”. This phrase needs to be moved to the M&M section. Line 202: figure 1. In the figure there are not the statistical variability index of data (i.e. SE). As suggested in the general comments, the data have to be analysed by regression analysis.

Reviewer 4 Report

please refer to the attaced file
